Pathway2Targets: an open-source pathway-based approach to repurpose therapeutic drugs and prioritize human targets

Dobbs Spendlove Mauri 1
M. Gibson Trenton 1
McCain Shaney 1
Stone Benjamin C. 1
Gill Tristan 2
http://orcid.org/0000-0001-7930-8160 Pickett Brett E. 1 brett_pickett@byu.edu
1 Microbiology and Molecular Biology, Brigham Young University , Provo, UT , United States of America
2 Carlsbad, California , United States
Ali Fares
Electronic publication date: 2023 Sep 29
Publication date: 2023
Volume: 11
Electronic Location ID: e16088
Received 2023 Mar 28; Accepted 2023 Aug 22
Copyright: © 2023 Dobbs Spendlove et al.
Copyright year: 2023
Copyright holder: Dobbs Spendlove et al.
License: This is an open access article distributed under the terms of the Creative Commons Attribution License, which permits unrestricted use, distribution, reproduction and adaptation in any medium and for any purpose provided that it is properly attributed. For attribution, the original author(s), title, publication source (PeerJ) and either DOI or URL of the article must be cited.
License URL: https://creativecommons.org/licenses/by/4.0/

Keywords: Drug repurposing, Drug targets, Target prioritization, Bioinformatics, Colorectal cancer, Target, Pathways, Prediction

Funding: The authors received no funding for this work.

==============================
Background

Recent efforts to repurpose existing drugs to different indications have been accompanied by a number of computational methods, which incorporate protein-protein interaction networks and signaling pathways, to aid with prioritizing existing targets and/or drugs. However, many of these existing methods are focused on integrating additional data that are only available for a small subset of diseases or conditions.

Methods

We have designed and implemented a new R-based open-source target prioritization and repurposing method that integrates both canonical intracellular signaling information from five public pathway databases and target information from public sources including OpenTargets.org. The Pathway2Targets algorithm takes a list of significant pathways as input, then retrieves and integrates public data for all targets within those pathways for a given condition. It also incorporates a weighting scheme that is customizable by the user to support a variety of use cases including target prioritization, drug repurposing, and identifying novel targets that are biologically relevant for a different indication.

Results

As a proof of concept, we applied this algorithm to a public colorectal cancer RNA-sequencing dataset with 144 case and control samples. Our analysis identified 430 targets and ~700 unique drugs based on differential gene expression and signaling pathway enrichment. We found that our highest-ranked predicted targets were significantly enriched in targets with FDA-approved therapeutics for colorectal cancer (p-value < 0.025) that included EGFR, VEGFA, and PTGS2. Interestingly, there was no statistically significant enrichment of targets for other cancers in this same list suggesting high specificity of the results. We also adjusted the weighting scheme to prioritize more novel targets for CRC. This second analysis revealed epidermal growth factor receptor (EGFR), phosphoinositide-3-kinase (PI3K), and two mitogen-activated protein kinases (MAPK14 and MAPK3). These observations suggest that our open-source method with a customizable weighting scheme can accurately prioritize targets that are specific and relevant to the disease or condition of interest, as well as targets that are at earlier stages of development. We anticipate that this method will complement other approaches to repurpose drugs for a variety of indications, which can contribute to the improvement of the quality of life and overall health of such patients.

Introduction

Substantial effort and resources have been devoted to identifying therapeutic treatments for many human diseases and conditions. The maladies could be caused by autoimmunity, uncontrolled cell growth, genetics, infection, and other acute or chronic ailments. Since moving a candidate treatment through the process of approval by the US Food and Drug Administration (FDA) is risky (Zhong et al., 2018), often taking many years, and requiring a substantial financial investment; researchers have expanded their development efforts to drug repurposing (Hernandez et al., 2017; Parvathaneni et al., 2019). Traditional methods of drug discovery have involved using low- or high-throughput screens to identify inhibitors or activators of a given target (Thakur et al., 2021; Olgen, 2019; Glanz et al., 2020). Hits that are identified in these screens are generally optimized prior to subsequent testing in cell culture, animal models, and clinical trials (Thakur et al., 2021). Alternatively, using a pathway-based approach to drug discovery involves performing experiments to better understand the underlying mechanism(s) of a given condition, and to identify relevant targets (Deng et al., 2020; Wang et al., 2020b; Damale et al., 2020; Chatterjee et al., 2022; Liu et al., 2021; Ren et al., 2010). Past studies have shown that incorporating a signaling pathway approach can successfully identify proteins that can be targeted with therapeutics having sufficient efficacy and safety to warrant approval by regulators (Ding et al., 2020; Khojasteh Poor et al., 2021; Choi et al., 2020; Proctor, Thompson & O’Bryant, 2014).

Drug repurposing is the process of getting regulatory approval for applying an existing therapeutic to a separate disease or condition (i.e., indication) (Ding et al., 2020; Zali et al., 2019; Harb, Lin & Hao, 2019). The benefits of this approach include a potentially shorter time to approval since the therapeutic has already been deemed as “safe” by government regulatory agencies. Early repurposing efforts were focused on identifying symptom similarities or using known side-effects from patients with other conditions to treat a separate condition (Kingsmore, Grammer & Lipsky, 2020; Ballard et al., 2020). Subsequent advances in understanding intracellular signaling mechanisms enabled a transition to more complex analyses that identify a candidate therapeutic for repurposing, and to develop novel therapeutics towards known targets (Schein, 2020). This is evidenced by the wide variety of drug and target discovery tools that have already been reported (Paananen & Fortino, 2020; Sleno & Emili, 2008; Huang et al., 2020). The majority of these modern methods take advantage of protein-protein interaction networks (Ma et al., 2019; Ozdemir et al., 2019; Cheng et al., 2019), gene sets (Masoudi-Sobhanzadeh et al., 2020; Tanoli, Vähä-Koskela & Aittokallio, 2021), and/or signaling pathways (Jain et al., 2021) for repurposing-based drug- and target prioritization efforts. Some methods combine one or more of these methods with artificial intelligence to further improve the pace of drug discovery (Tanoli, Vähä-Koskela & Aittokallio, 2021; Anderson et al., 2020; Paul et al., 2021; Gupta et al., 2021).

Even with such recent advances, many prioritization algorithms rely on public or proprietary protein network data (Emig et al., 2013; Huang et al., 2014; Louhimo et al., 2016; Li & Lu, 2013; Isik et al., 2015; Carrella et al., 2014; Setoain et al., 2015; Duan et al., 2016; Barrio-Hernandez et al., 2023; Fang et al., 2019; Lee et al., 2011; Greene et al., 2015; Huang et al., 2018), with some algorithms focusing on a particular set of diseases or conditions (Crowther et al., 2010; Xu, Kong & Hu, 2021; Dezső & Ceccarelli, 2020; Fiscon et al., 2021; Chen & Xu, 2016; Regan-Fendt et al., 2020). Drug repurposing and target prioritization algorithms generally apply a consistent set of parameters, which are often specific to a given indication. Such specialization makes it difficult to effectively and adequately support the efforts of researchers working in other disease areas (Sharma et al., 2021; Begley et al., 2021).

Given the specialization that is prevalent among many repurposing tools, the aim of the current study was to incorporate a novel, flexible, and customizable open-source target prioritization method into the Pathway2Targets algorithm, which would increase the number of supported use cases. This updated algorithm retrieves additional target information, clinical trial data, automatically fetches the reactome pathway diagrams for the signaling pathways with the highest number of targets, and accepts reactome pathway enrichments generated by the enrichr algorithm (Xie et al., 2021a). This additional data and prioritization method are used by the updated algorithm to generate ranked lists of targets and therapeutics that can be applicable to multiple use cases (Scott, Jensen & Pickett, 2021; Gray et al., 2022; Moreno et al., 2022; Rapier-Sharman, Clancy & Pickett, 2022). The entities in these lists can then be evaluated as candidates for condition-specific repurposing efforts based solely on the unique signaling pathway “profile” for the disease/condition of interest.

Method

GEO query for transcriptomics data

The Gene Expression Omnibus database was queried for a well-controlled bulk transcriptomics human colorectal cancer dataset with a sufficiently high number of samples (GSE156451) to enable confident downstream repurposing analysis (Sayers et al., 2021). The paired-end fastq files for this publicly available study were then downloaded from the Sequence Read Archive (SRA), a database within the National Center for Biotechnology Information (Sayers et al., 2021). This study consisted of 144 samples, with 72 from tumors in patients with colorectal cancer (CRC) and the other 72 from native human tissue (Li et al., 2021).

Transcriptomic preprocessing and analysis

The 144 public colorectal cancer transcriptomics samples were preprocessed using the Automated Reproducible MOdular Workflow for Preprocessing and Differential Analysis of RNA-seq Data (ARMOR) software (Orjuela et al., 2019). Briefly, this open-source Snakemake-based workflow was used to perform read trimming on the RNA-sequencing fastq files with TrimGalore (Köster & Rahmann, 2018; https://www.bioinformatics.babraham.ac.uk/projects/trim_galore/), determine quality control metrics with FastQC (www.bioinformatics.babraham.ac.uk/projects/fastqc/), map and quantify reads to the human GRCh38 transcriptome with Salmon (Patro et al., 2017), and calculate differential gene expression with edgeR (Robinson, McCarthy & Smyth, 2010) by comparing the CRC samples (case) to the native tissue samples (control). The Ensembl Gene IDs that were generated by the edgeR algorithm were converted to Entrez Gene IDs using a R-based application programming interface (API) to the BiomaRt database prior to pathway analysis (Kasprzyk, 2011). Similarly, the enrichr pathway enrichment software only required a gene symbol, log2 fold-change values, and FDR-adjusted p-values as input from the DEG list. The statistically significant differentially expressed genes (DEGs; FDR-corrected p-value < 0.05) were then subjected to signaling pathway analysis using the Signal Pathway Impact Analysis (SPIA) algorithm with 3,000 bootstrap replicates to generate a null distribution for each of over 2,000 public signaling pathways (Tarca et al., 2009), as reported previously (Scott, Jensen & Pickett, 2021; Gray et al., 2022; Moreno et al., 2022; Rapier-Sharman, Clancy & Pickett, 2022; Ferrarini et al., 2021; Scott et al., 2022; Gifford & Pickett, 2022). The lists of pathways were derived from publicly available versions of KEGG (Aoki-Kinoshita & Kanehisa, 2007), Reactome (Jassal et al., 2020), Pathway Interaction Database (Schaefer et al., 2009), BioCarta, and Panther (Mi et al., 2017).

Target data acquisition and integration

The only input file required for the Pathway2Targets software was the tabular output file containing the significant signaling pathways (Bonferroni-corrected p-value < 0.05) generated by SPIA, although an output file from the enrichr algorithm would have also been compatible (Xie et al., 2021a). The Pathway2Targets software then programmatically retrieved the gene products that were members of each significant pathway from the five pathway databases mentioned previously, and then obtained the UniProt protein identifiers for each Ensembl ID using the BiomaRt API (Scott, Jensen & Pickett, 2021; UniProt Consortium, 2019). A GraphQL query was automatically generated and submitted through the Open Targets Platform API to access the relevant drug and target information for each of the UniProt protein identifiers in each pathway (Ochoa et al., 2021). These additional data for each target included the number of associated diseases, tractability, subcellular location, safety, number of unique drugs, number of signaling pathways, number of FDA-approved therapeutics, number of therapeutics in phase-three clinical trials, number of therapeutics in phase-two clinical trials, number of therapeutics in phase-one clinical trials, and number of therapeutics in phase-four clinical trials. This information was then automatically integrated with the significant results from the signaling pathway enrichment analysis described above in a single table to facilitate downstream target scoring and prioritization.

Target weighting factors

A logical and customizable weighting scheme was constructed in the algorithm that would compile and analyze the data for all existing therapeutics for each pathway member to facilitate target prioritization (Table 1). The default weights for each target attribute were specifically chosen in a way that would prioritize targets present in multiple pathways, a high number of associated disease, and a higher number of therapeutics further along in clinical trials. However, these default weighted values could also be easily adjusted to customize the output based on individual prioritization preferences or desired outcome. As a proof of concept for adjusting the default weighted values, these values were adjusted to facilitate discovery of novel and/or early-stage targets (Table 1). A table of prioritized targets as well as a separate table of prioritized therapeutics, which uses the same target prioritization weighting scheme was then generated as output from the Pathway2Targets software. An option to automatically download the graphical representations for the most common Reactome pathways, via the Reactome API, was also provided (Jassal et al., 2020).

Table 1 Attributes and weighting values used to prioritize targets for repurposing (default) and for early/novel target prediction.

Target attribute	Default weighted score	Weighted score for novel targets	
Number targets in pathways	1	0.5	
Tractability	1	0.5	
Number approved drugs	1	0.5	
Number safety liabilities	−2	−2	
Number unique drugs	1	0.5	
Number associated diseases	1	0.01	
Phase 1 drug	0.5	10	
Phase 2 drug	1	4	
Phase 3 drug	1.5	0.5	
Phase 4 drug	2	0.01	

Results

Original software implementation

Our original implementation of the Pathway2Targets algorithm, implemented in R, required only a single input file containing the output from a SPIA-based signaling pathway enrichment analysis. This original version of the Pathway2Targets algorithm then iterated over each significant signaling pathway name (p-value < 0.05), together with the associated source databases, and retrieved all gene products that are members of each pathway. The diverse public identifiers used by each pathway database were then automatically converted and submitted to the Open Targets database to retrieve a smaller subset of the available target information. The data for ~10 fields associated with each target were then programmatically retrieved and stored for processing. These fields included target name, target symbol, target Ensembl identifier, therapeutic name, therapeutic type, modulation of the target, total tractability, maximum clinical phase, and approval status. The single output file combined these fields with the relevant pathway information for each target (Table S1). Unfortunately, we found that the file generated by the original software required substantial downstream manual review and analysis to facilitate biologically relevant interpretation. Importantly, while the earlier version of the software did identify relevant targets from the significant signaling pathways, it did not collect sufficient data from the Open Targets database to enable the prioritization of targets directly from the output.

Novel software enhancements

In order to address the observed inefficiencies in the original algorithm and to facilitate improved target prioritization, we made substantial enhancements to the original Pathway2Targets algorithm. The updated software still only requires a tabular input file containing the significant signaling pathways. However, the updated version now retrieves data for ~20 additional fields from the Open Targets Platform, calculates additional metrics, and incorporates a customizable prioritization weighting scheme (Fig. 1). Specifically, we updated the software to retrieve additional Open Targets data including tractability for each therapeutic modality, subcellular location, number of unique drugs, number of signaling pathways, number of therapeutics in phase-three clinical trials, number of therapeutics in phase-two clinical trials, number of therapeutics in phase-one clinical trials, and number of therapeutics in phase-four clinical trials. We also updated the software to calculate the number of targets that are present in each pathway.

Figure 1 Diagram representing Pathway2Targets workflow, various data sources, and outputs.

Upstream processing steps include differential gene expression and signaling pathway enrichment analysis. The algorithm uses significant pathways and target data from the open targets platform to perform drug- and target prioritization.

These enhancements to Pathway2Targets enabled us to incorporate a set of logical, flexible, and customizable weighting factors that directly support target prioritization efforts. To do so, we assigned customizable weights to each of 10 target attributes that are commonly used by researchers to rank/prioritize pathway-based targets (Table 1). We also implemented new functionality to automatically retrieve pathway diagrams from Reactome and to support target prioritization efforts by using the enrichr algorithm on the Reactome database.

Application to a colorectal cancer use case: gene expression & pathways

After implementing these enhancements into Pathway2Targets, we next decided to validate the biological relevance of the results from this algorithm. To do so, we tested it on the significantly enriched signaling pathways from the public RNA-sequencing dataset in colorectal cancer. To perform this analysis, we downloaded and preprocessed the fastq files for this dataset using the ARMOR automated workflow that performed quality control, trimmed and mapped reads, and calculated 12,159 differentially expressed genes (FDR-adjusted p-value < 0.05) (Fig. 2; Table S2). Examples of the most significant DEGs included KRT80, AJUBA, TRIP13, PAICS, and RNASEH2A with FDR-adjusted p-values ranging from 1 × 10−37 to 1 × 10−34. Significant DEGs that were up-regulated include MMP7, PPBP, and CXCL5; while a subset of down-regulated DEGs included OTOP3, OTOP2, SPIB, and INSL5 (Table 2; Table S3).

Figure 2 A volcano plot of the differentially expressed genes (DEGs) calculated from tumors from patients with colorectal cancers compared to native patient tissue.

Dots represent individual genes and colors indicate down-regulation, up-regulation, or expression that did not surpass the threshold of |log2 fold-change| < 1 (blue, red, and gray respectively).

Table 2 The top 20 differentially expressed genes, ranked in descending order by the absolute value of the log2 fold-change values.

Gene symbol	Gene description	Ensembl gene ID	Entrez gene ID	log2 fold-change	FDR p-value	
PPBP	Pro-platelet basic protein	ENSG00000163736	5473	7.23	8.75E−19	
MMP7	Matrix metallopeptidase 7	ENSG00000137673	4316	5.91	1.11E−28	
KLK6	Kallikrein related peptidase 6	ENSG00000167755	5653	5.76	2.35E−21	
CA9	Carbonic anhydrase 9	ENSG00000107159	768	5.75	8.76E−25	
OTOP3	Otopetrin 3	ENSG00000182938	347741	−5.68	4.20E−28	
DHRS2	Dehydrogenase/reductase 2	ENSG00000100867	10202	5.61	1.43E−18	
CXCL5	C-X-C motif chemokine ligand 5	ENSG00000163735	6374	5.49	1.95E−21	
REG1A	Regenerating family member 1 alpha	ENSG00000115386	5967	5.33	7.47E−15	
NOTUM	Notum, palmitoleoyl-protein carboxylesterase	ENSG00000185269	147111	5.32	1.21E−16	
CST1	Cystatin SN	ENSG00000170373	1469	5.1	2.80E−26	
CPNE7	Copine 7	ENSG00000178773	27132	5.08	3.38E−28	
SLC35D3	Solute carrier family 35 member D3	ENSG00000182747	340146	5.06	5.77E−26	
OTOP2	Otopetrin 2	ENSG00000183034	92736	−4.95	5.27E−21	
KRT6B	Keratin 6B	ENSG00000185479	3854	4.9	4.76E−17	
CLDN2	Claudin 2	ENSG00000165376	9075	4.84	2.34E−20	
COL11A1	Collagen type XI alpha 1 chain	ENSG00000060718	1301	4.79	1.07E−23	
KRT80	Keratin 80	ENSG00000167767	144501	4.75	1.05E−37	
BEST4	Bestrophin 4	ENSG00000142959	266675	−4.73	3.22E−23	
FOXQ1	Forkhead box Q1	ENSG00000164379	94234	4.71	1.21E−24	
TACSTD2	Tumor associated calcium signal transducer 2	ENSG00000184292	4070	4.7	4.96E−27	

To identify significant signaling pathways in this public colorectal cancer dataset we then applied the existing signaling pathway impact analysis (SPIA) algorithm to the list of DEGs in CRC. The SPIA algorithm uses permutation and bootstrapping to generate a null distribution for each pathway, and then applies a Bonferroni p-value correction to reduce the number of false-positive results. This pathway enrichment analysis identified 63 statistically significant (Bonferroni-adjusted p-value < 0.05) and biologically relevant intracellular signaling pathways including the inhibition of HIF-1 (Bonferroni p-value 0.0017), activation of cell cycle (Bonferroni p-value 0.003), activation of DNA replication (Bonferroni p-value 0.0008), activation of PLK1 (Bonferroni p-value 0.000089), and inhibition of T-cell receptor signaling pathways (Bonferroni p-value 0.001) (Table 3).

Table 3 Top 10 most significant intracellular signaling pathways that are predicted to be affected in colorectal cancer.

Name	# Proteins in pathway	# DEGs in pathway	Bonferroni-adjusted p-value	Pathway status	Source database	
Major pathway of rRNA processing in the nucleolus and cytosol	169	162	7.19E−13	Activated	Reactome	
rRNA processing in the nucleus and cytosol	178	169	3.55E−12	Activated	Reactome	
rRNA processing	183	173	6.45E−12	Activated	Reactome	
Gene expression	1,598	1,307	3.45E−08	Activated	Reactome	
Eukaryotic translation elongation	90	86	7.11E−08	Activated	Reactome	
Peptide chain elongation	86	82	1.64E−07	Activated	Reactome	
Processing of capped intron-containing pre-mRNA	228	202	2.61E−07	Activated	Reactome	
Translation	152	137	1.01E−06	Activated	Reactome	
Non-coding RNA metabolism	49	48	1.86E−06	Activated	Reactome	
snRNP assembly	49	48	1.86E−06	Activated	Reactome	

Application to a colorectal cancer use case: late-stage target prioritization from signaling pathways

After identifying significantly affected signaling pathways, we then wanted to determine whether our updated Pathway2Targets software could be used for two purposes: (1) to predict existing late-stage therapeutics from the enriched signaling pathway data, as well as those that could potentially be repurposed for treating colorectal cancer and (2) to generate a ranked list of prioritized targets that were specific to colorectal cancer. To evaluate these outcomes, we used the new Pathway2Targets software to analyze our significant pathway results from the CRC dataset.

Our Pathway2Targets analysis generated a list of 430 targets (Table S4) and approximately 700 unique drugs (Table S5). To determine which signaling pathways were enriched in the list of targets, we ran an enrichment in Reactome and visualized the result for the “Oxygen-dependent proline hydroxylation of Hypoxia-inducible Factor Alpha” Reactome pathway (Fig. 3).

Figure 3 The “Oxygen-dependent proline hydroxylation of Hypoxia-inducible Factor Alpha” signaling pathway from Reactome.

Portions of rectangular nodes are shaded yellow to represent the fraction of components within each node that were included identified as targets in our analysis.

Application to a colorectal cancer use case: significant enrichment of targets for indication

We next wanted to quantify how many of the predicted targets were approved for either colorectal cancer and/or any cancer. To do so, we manually reviewed the top 50 targets, together with the attributes and default weighted scores. This analysis showed that the top 50 prioritized targets, using the default weighting parameters, included three approved CRC targets and 11 targets for other cancers. Given that many labs prefer to validate a much smaller number of results in the wet-lab, we reduced the top 50 to a list of 15 potential candidates for drug repurposing. After applying this filtering process, we observed three CRC targets (VEGFA, EGFR, and PTGS2) and five targets that had undergone one or more phase IV trials in any indication (TP53, VEGFA, EGFR, ESR1, and PTGS2) were all found in the top 15 results from this list (Table 4).

Table 4 Ranked list of attributes for three approved colorectal cancer drug targets, predicted using our weighted factors.

	Target symbol	
	VEGFA	EGFR	PTGS2	
Target ID	ENSG00000112715	ENSG00000146648	ENSG00000073756	
Target name	Vascular endothelial growth factor A	Epidermal growth factor receptor	Prostaglandin-endoperoxide synthase 2	
Associated disease count	2,188	1,839	1,585	
Tractability count	5	7	4	
sm (is approved)	FALSE	TRUE	TRUE	
sm (is in advanced trial)	FALSE	FALSE	FALSE	
sm (is in phase 1)	FALSE	FALSE	FALSE	
ab (is approved)	FALSE	TRUE	TRUE	
ab (is in advanced trial)	TRUE	TRUE	FALSE	
ab (is in phase 1)	FALSE	FALSE	FALSE	
pr (is approved)	FALSE	FALSE	FALSE	
pr (is in advanced trial)	FALSE	FALSE	FALSE	
pr (is in phase 1)	FALSE	FALSE	FALSE	
oc (is approved)	TRUE	FALSE	TRUE	
oc (is in advanced trial)	FALSE	TRUE	FALSE	
oc (is in phase 1)	FALSE	FALSE	FALSE	
Subcellular location	Secreted	Cell membrane	Microsome membrane	
Safety liabilities	0	2	25	
Number unique drugs	11	72	75	
# Pathways with target	2	1	1	
# Approved therapeutics	4	14	12	
# Therapeutics in phase 3	1	2	0	
# Therapeutics in phase 2	0	0	0	
# Therapeutics in phase 1	0	0	0	
# Therapeutics in phase 4	4	14	12	
Weighted score	2,219.5	1,960	1,651	
Prioritized rank	3	6	12	
Note:

sm, small molecule; ab, antibody; pr, proteolysis targeting chimeras; oc, other clinical modalities.

To determine whether our selected parameters were able to accurately predict therapeutic candidates that would be specific for colorectal cancer or all cancers, we performed a hypergeometric statistical analysis of the top 15 targets. This analysis showed a significant enrichment of CRC targets in the top 15 predicted results (p-value < 0.025), while there was no significant enrichment for all cancers (p-value = 0.13). This result suggests that our approach is capable of predicting biologically relevant targets based on the specific signaling pathway profile of the disease/system being evaluated.

Application to a colorectal cancer use case: early-stage target prioritization from signaling pathways

Since the flexible and customizable design of our updated software is capable of supporting multiple use cases, we next adjusted the weighting scheme to mimic a scenario where the default weights would be adjusted to enable the prediction of early-stage or novel targets. To achieve such a scenario, we decreased the weight for targets in pathways, tractability count, number of approved drugs, number of unique drugs, and phase 3 from the default value of 1 to a new value of 0.5. We also reduced the weight for both associated diseases and phase 4 from the default of 1 to 0.01, increased phase 2 from the default of 1 to 4, and increased phase 1 from the default of 0.5 to 10. The results of this analysis showed that EGFR was the only CRC-approved target that remained in the top-10 (Table 5), while VEGFA and PTGS2 decreased in rank (to 41 and 425 respectively) (Tables S6, S7). Interestingly, this modified weighting scheme resulted in MTOR, MAPK14, and TP53 being present in the top 10 results. This observation suggests that our customizable weighting approach could be capable of predicting relevant potential drug repurposing candidates even when minimal approved targets are known.

Table 5 The top 10 early/novel targets in colorectal cancer, based on the customized weights defined in Table 1.

Target symbol	Target name	Subcellular location	Weighted score	
EGFR	Epidermal growth factor receptor	Cell membrane	61.43	
MAPK14	Mitogen-activated protein kinase 14	Cytoplasm	60.12	
RPS6KB1	Ribosomal protein S6 kinase B1	Synapse	59.89	
MTOR	Mechanistic target of rapamycin kinase	Endoplasmic reticulum membrane	54.94	
PIK3R1	Phosphoinositide-3-kinase regulatory subunit 1	Cytosol	54.53	
TP53	Tumor protein p53	Cytoplasm	52.83	
AURKB	Aurora kinase B	Nucleus	50.9	
PIK3R2	Phosphoinositide-3-kinase regulatory subunit 2	No data	49.29	
CD40	CD40 molecule	Cell membrane	48.63	
MAPK3	Mitogen-activated protein kinase 3	Cytoplasm	48.6	

Discussion

This study reports the capabilities of the updated Pathway2Targets algorithm to predict, prioritize, and/or repurpose targets for a particular disease or condition by combining pathway information with additional public data and a weighting scheme in a flexible and customizable way. This approach improves on our prior work to incorporate the significant intracellular signaling pathways that best represent the underlying transcriptomic differential expression data into the target prediction and prioritization (Scott, Jensen & Pickett, 2021; Gray et al., 2022; Moreno et al., 2022; Rapier-Sharman, Clancy & Pickett, 2022; Ferrarini et al., 2021; Scott et al., 2022; Gifford & Pickett, 2022). Our target prediction and prioritization approach relies on the underlying significant mechanistic pathways, based on large amounts of -omics data. This design makes our software disease-agnostic, and capable of prioritizing targets across a wide range of diseases based solely on the signaling pathway profile. In brief, our algorithm accurately predicted three approved targets with high specificity for CRC, using only signaling pathway results, relevant target information, and the default weighting scheme. It was also able to prioritize relatively early-stage and novel targets for CRC by adjusting the weight parameters. These findings suggest that our pathway-based repurposing approach could be relevant for other oncological, rare disease, or other indications. Such repurposing could potentially predict therapeutics that could still work for a given indication but that may be more economical-thereby reducing the cost of treatment in developing regions of the world.

An extensive array of drug- and target prioritization approaches have been developed and reported previously. In particular, some algorithms specifically focus on performing these tasks for particular diseases or conditions (Di et al., 2019; Dwane et al., 2021; Yang et al., 2021b; Behan et al., 2019; Mejía-Pedroza, Espinal-Enríquez & Hernández-Lemus, 2018; Urán Landaburu et al., 2020; Tsuji et al., 2021). Many other implemented methods are informed by protein-protein interaction networks and/or pathway information for either target prioritization or drug repurposing (Ma et al., 2019; Isik et al., 2015; Fiscon & Paci, 2021; Aguirre-Plans et al., 2019; Napolitano et al., 2018; Wang et al., 2019a; Malas et al., 2019), but are not designed to perform both tasks simultaneously as ours does. A subset of these algorithms even incorporate machine learning to improve their results (Tsuji et al., 2021; Malas et al., 2019). Our approach differs from these prior tools by only requiring a list of enriched signaling pathways from either SPIA or enrichr as input, implementing a flexible weighting scheme for target prioritization, and should therefore be more broadly applicable to diverse indications.

As a proof-of-concept, our secondary analysis of publicly available colorectal cancer data unsurprisingly identified thousands of DEGs. While the signaling pathways that were significantly enriched by these DEGs were used as input to the Pathway2Targets algorithm, we believe it is important to validate the findings that we observed upstream of the signaling pathway data. Some of the most highly up-regulated gene products in our analysis included MMP7, PPBP, and CXCL5. Prior work by others has identified the MMP7 (Gao et al., 2019; Huang et al., 2021; Vočka et al., 2019), PPBP (Chen et al., 2022; Kothalawala & Győrffy, 2023; Feng et al., 2022), and CXCL5 gene products to be extremely useful in the mechanisms and diagnosis of colorectal cancer (Chen et al., 2019; Zhang et al., 2021; Novillo et al., 2020; Zhang et al., 2020). We also identified multiple down-regulated genes in colorectal cancer that are supported by recent studies on the gene products of the OTOP2 (Qu et al., 2019; Yang & Sakharkar, 2022), OTOP3 (Yang & Sakharkar, 2022), SPIB (Zhao et al., 2021; Liu et al., 2019), and INSL5 genes (Yang et al., 2021c; Sun et al., 2019). Similarly, a subset of the signaling pathways that we observed have previously been shown to be relevant for this indication including “HIF-1 signaling pathway” (Lamberti et al., 2019; Seo et al., 2021) and “PLK1 signaling events” (Yu et al., 2021; Xie et al., 2021b).

We predicted the targets from our Pathway2Targets algorithm using only the significant signaling pathways as input, with all other required data retrieved programmatically. Although our initial target prioritization analysis identified hundreds of targets, many researchers tend to focus only on the highest-ranking results for follow-up validation work. Our observation that the top 15 highest-scoring targets were significantly enriched in approved drugs (EGFR, VEGFA, and PTGS2) demonstrates that our flexible weighting scheme can accurately focus subsequent wet-lab validation on “hits” that are more likely to be effective, as has been done previously (Gray et al., 2022; Moreno et al., 2022; Rapier-Sharman, Clancy & Pickett, 2022; Zeng et al., 2020; Madhukar et al., 2019). It has not escaped our notice that many of the highest scoring prioritized targets predicted by our algorithm are also well-established biomarkers of cancer including p53, IL-6, VEGFA, EGFR, NF-KB, ESR1, ERBB2, and others (Pentheroudakis et al., 2019; Wang et al., 2020a; Liebl & Hofmann, 2021; Vainer, Dehlendorff & Johansen, 2018; Wang et al., 2019b; Li et al., 2020; Lupo et al., 2020; Ye et al., 2020; Topi et al., 2022; Bitar et al., 2021).

Our second target prioritization analysis simulated a scenario that involved prioritizing targets when no/minimal approved drugs exist. Our customizable weighting scheme made this possible by quickly changing the weight parameters in the software. Interestingly, the results from this analysis identified many targets that are relevant to oncology-related indications including MTOR (Hillmann & Fabbro, 2019; du Rusquec et al., 2020; Lengyel et al., 2020), MAPK14 (Fang & Richardson, 2005; Chen et al., 2013; Wang et al., 2022), and TP53 (Jovanović et al., 2018; Ciccarese et al., 2017; Yang et al., 2021a). Each of these targets have multiple therapeutics at early stages in the pipeline, and consequently rose to the top of the list. This suggests that our approach can accurately identify significant signaling pathways that represent a profile that is both unique to a given disease/condition as well as overlapping with relevant diseases/conditions.

The novelty of our Pathway2Targets target prediction algorithm consists of multiple factors such as open-source code, a single input file of significant pathways, a flexible weighting scheme, incorporation of signaling pathways, output of prioritized targets and therapeutics for repurposing open combines significant signaling pathway enrichment results with publicly available data and our custom weighting scheme can be integrated with safety and efficacy data to further complement ongoing prioritization and repurposing efforts. While the default weighting scheme appears to be capable of identifying relevant targets for CRC in our first analysis, the default parameters may not be universally applicable to all possible use cases. We envision that modifying the weighting scheme should adequately support a wider set of use cases.

We believe that these predictions of approved and novel targets for CRC show promise and that the software warrants additional investigation in other indications. Additional work will be required to determine the extent to which this software can be applied to other -omics technologies such as whole genome sequencing, chromatin immunoprecipitation sequencing (ChIP-seq), shotgun proteomics, etc. It is imperative that the user provides high-quality data with minimal numbers of confounding variables, such as timepoints, as input. Although our predictions for CRC are enriched in approved targets, it is important to note that any target predictions made by this software should be confirmed in well-controlled registered clinical trials prior to use in the clinic. Users of the software should also keep in mind that the algorithm only identifies existing therapeutics that are present in the Open Targets platform. As such, it is unrealistic to expect the software to predict drugs that have not been publicly disclosed or that it performs docking predictions to identify novel molecules that bind to novel targets. Similarly, those applying the results to early-stage research should account for toxicity or off-target effects that are associated with a given therapeutic. Even with these caveats, we believe our algorithm can be applied across a broad variety of human diseases and conditions. To do so, users would only need to generate a list of DEGs from their disease of interest, run a pathway enrichment analysis using either SPIA or enrichr, and then use the pathway results as input to the Pathway2Targets software. Future collaborative work will be required to determine how Pathway2Targets can be integrated with precision medicine, artificial intelligence, or other approaches.

We expect that researchers will be interested in applying this open-source target prioritization algorithm to predict the most relevant targets for a disease space regardless of competition in the field. However, some users may need to modify the default weights for one or more parameters in order to yield improved results for any given use case. Although additional future analyses are needed to confirm that this approach remains relevant in other human diseases and conditions, we expect that our flexible and customizable weighting approach will facilitate such efforts from gene expression and pathway data. In conclusion, we expect that this improved algorithm will facilitate future predictions of therapeutic targets that could be repurposed for other indications.

Operating system

This software is platform-independent and has been successfully tested on 64-bit RedHat Linux and on Mac OS 12.0 and 13.0.

Programming language

This software is written in R (version 3.6.1 or later).

Supplemental Information

Supplemental Information 1 Target-related results generated by the earlier version of Pathway2Targets.

Click here for additional data file.

Supplemental Information 2 Differentially expressed genes (DEGs) identified by re-analyzing the public colorectal cancer dataset.

Click here for additional data file.

Supplemental Information 3 List of significant signaling pathways that are associated with colorectal cancer.

Click here for additional data file.

Supplemental Information 4 List of targets prioritized with default weighting values.

Click here for additional data file.

Supplemental Information 5 List of therapeutics for targets prioritized with default weighting values.

Click here for additional data file.

Supplemental Information 6 List of targets prioritized with adjusted weighting values to enable targets with drugs at earlier stages of development.

Click here for additional data file.

Supplemental Information 7 List of therapeutics for targets prioritized with adjusted weighting values to enable targets with drugs at earlier stages of development.

Click here for additional data file.

We gratefully acknowledge the Office of Research Computing at Brigham Young University for providing the computational infrastructure, training, and technical support needed to successfully complete this work.

Additional Information and Declarations

Competing Interests

Author Contributions

Data Availability

Brett E. Pickett is an Academic Editor for PeerJ.

Mauri Dobbs Spendlove performed the experiments, analyzed the data, authored or reviewed drafts of the article, and approved the final draft.

Trenton M. Gibson performed the experiments, analyzed the data, authored or reviewed drafts of the article, and approved the final draft.

Shaney McCain performed the experiments, analyzed the data, authored or reviewed drafts of the article, and approved the final draft.

Benjamin C. Stone performed the experiments, analyzed the data, authored or reviewed drafts of the article, and approved the final draft.

Tristan Gill conceived and designed the experiments, authored or reviewed drafts of the article, and approved the final draft.

Brett E. Pickett conceived and designed the experiments, performed the experiments, analyzed the data, prepared figures and/or tables, authored or reviewed drafts of the article, and approved the final draft.

The following information was supplied regarding data availability:

This software is available at GitHub and Zenodo:

- https://github.com/bpickett/Pathway2Targets.

- bpickett. (2023). bpickett/Pathway2Targets: v2.26 (v2.26). Zenodo. https://doi.org/10.5281/zenodo.7693815.

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
