# Peer review of "Pathway2Targets: an open-source pathway-based approach to repurpose therapeutic drugs and prioritize human targets"

_PeerJ, doi:10.7717/peerj.16088_

## Round 0.1 · original submission · Major Revisions

All the reviewer's recommendations must be carefully addressed as indicated.

Reviewer 1 ·

Basic reporting

Spendlove et al established a new open-source prioritization and repurposing approach which can be used in target prioritization, signaling pathway identification, as well as drug repurposing. The authors tried to validate the biological relevance of their new algorithm using colorectal cancer as the example. In Fig.2, the authors highlighted several DEGs that being identified from their algorithm. However, they didn’t provide rationale on why these genes were highlighted. Apparently, they were not selected according to the fold change. In addition, it is highly recommended to include a table containing top 10 or 20 most upregulated and downregulated genes along with Figure 2 for better clarification.

Experimental design

The authors introduced new algorithms to provide improvement on the previous software. However, the authors need to provide evidence clearly showing such enhancement do provide better readout compared to the previous version of the software. For instance, they need to run the same dataset using both old and new version of their software and make comparisons between the two.

Validity of the findings

The authors also tried to identify novel therapeutic targets and the results from Supplementary Table 4&5 look very exciting and promising. This is actually the most valuable data in this manuscript and needs to be included not only in the Supplements but also in the main context. It is highly recommended to generate a new Figure 4, containing their modified weighting value table as well as the top 10 candidate genes from their analysis.

Additional comments

Please correct the typo in line 209, should be EGFR not EGRF.

Reviewer 2 ·

Basic reporting

Authors have reported the An open-source pathway-based approach to repurpose therapeutic drugs and prioritize human targets. I examined the title quite intreating and relevant. The authors demonstrated fine outcomes from the present study. The experimental design was well established with proper relevant parameters and data. The entire manuscript is looking concise and simple to understand but still, it has some significant margin for improvisation, after necessary corrections the present manuscript will be entitled to further consideration. Following rooms for corrections and responses to questions:

Experimental design

What was the prior version of the target prioritization algorithm, and how did the new algorithm improve upon it?

Author can explain how the new algorithm is flexible and customizable, and what are some of the use cases it can support.

How were the 144 samples used in the study collected and processed, and what types of data were analyzed?

What were the main findings of the study, and how do they relate to the field of cancer research?

Were there any limitations or challenges encountered during the study, and how were they addressed?

How might the findings of this study be translated into clinical practice, and what implications might they have for the development of new cancer therapies?

How might this algorithm be used in conjunction with other types of genomic data, such as gene expression or epigenetic data, to further improve target prioritization for cancer therapy?

How might this algorithm be adapted or applied to other types of cancer or diseases beyond colorectal cancer?

Validity of the findings

Author should explain how does a pathway-based approach differ from traditional drug discovery approaches, and what are the advantages and disadvantages of this approach?

Author can provide some examples of successful drug repurposing efforts that have used a pathway-based approach?

How can an open-source approach to drug repurposing and target prioritization facilitate collaboration and knowledge-sharing among researchers?


How can this approach be used to address rare diseases and conditions that may not receive as much attention from the pharmaceutical industry?

Are there any ethical or regulatory considerations that need to be taken into account when repurposing drugs for new indications using this approach?

What are the implications of this approach for the cost and availability of drugs for patients, particularly in low-income countries or regions with limited healthcare resources?

How can this approach be used in conjunction with other emerging technologies, such as artificial intelligence and precision medicine, to further accelerate drug discovery and development?

Additional comments

Kindly check carefully the grammatical and some spelling error in the manuscript that needs editing for publication.

---

## Round 0.2 · Minor Revisions

The reviewer comments should be addressed as recommended.

Reviewer 1 ·

Basic reporting

No comment

Experimental design

No comment

Validity of the findings

No comment

Reviewer 2 ·

Basic reporting

no comment

Experimental design

The authors clearly answered the experimental technical comments raised by the reviewers. While they did some effort to improve the writing style, the authors are recommended to again review the paper and do the necessary modifications when found, especially the scientific structure.

Validity of the findings

no comment

---

## Round 0.3 · accepted · Accept

The manuscript is now suitable for publication in PeerJ.

Reviewer 2 ·

Basic reporting

No comment

Experimental design

No comment

Validity of the findings

No comment